# CigTime: Corrective Instruction Generation Through Inverse Motion Editing

Qihang Fang[1], Chengcheng Tang[2], Bugra Tekin[2], and Yanchao Yang[1*]

[1]The University of Hong Kong
[2]Meta Reality Labs
{qihfang}@gmail.com, {chengcheng.tang,bugratekin}@meta.com, {yanchaoy}@hku.hk

## Abstract

Recent advancements in models linking natural language with human motions have shown significant promise in motion generation and editing based on instructional text. Motivated by applications in sports coaching and motor skill learning, we investigate the inverse problem: generating corrective instructional text, leveraging motion editing and generation models. We introduce a novel approach that, given a user's current motion (source) and the desired motion (target), generates text instructions to guide the user towards achieving the target motion. We leverage large language models to generate corrective texts and utilize existing motion generation and editing frameworks to compile datasets of triplets (source motion, target motion, and corrective text). Using this data, we propose a new motion-language model for generating corrective instructions. We present both qualitative and quantitative results across a diverse range of applications that largely improve upon baselines. Our approach demonstrates its effectiveness in instructional scenarios, offering text-based guidance to correct and enhance user performance.

## 1 Introduction

Corrective instructions are crucial for learning motor skills, such as sports. Without feedback, people are at risk of developing improper, suboptimal, and injury-prone moves that hinder long-term progress and health. With the growing popularity and immersion of motion-sensing sports games, the increasing accuracy and accessibility of 3D pose estimation techniques, and the advancement of fitness equipment and trackers with versatile sensing technologies, the need for intelligent coaching systems that provide corrective feedback on user motion is becoming increasingly important.

In this work, we study the task of *Motion Corrective Instruction Generation*, which aims to create text-based guidance to help users correct and improve their physical movements. This task has significant applications in sports coaching, rehabilitation, and general motor skill learning, providing users with precise and actionable instructions to enhance their performance. By leveraging advancements in human motion generation and editing, this task addresses the need for personalized and adaptive feedback in various instructional scenarios.

Recent research in text-conditioned human motion generation has shown impressive progress. Methods like MotionCLIP [39] and TEMOS [34] have utilized neural networks and transformer-based models to align text and motion into a joint embedding space, producing diverse and high-quality motion sequences. These models, however, focus primarily on generating motions from text rather than generating corrective instructions from motion pairs. Therefore they are not directly suitable for analyzing and improving user movements based on a comparison of motion sequences.

38th Conference on Neural Information Processing Systems (NeurIPS 2024).

Research specifically focused on corrective instruction generation is still in its early stages. Traditional methods often rely on building statistical models for specific action categories, which require expert experience and are difficult to scale and generalize to various actions. For example, Pose Trainer [6] and AIFit [11] employ neural networks and statistical models to provide feedback on specific exercises, but these methods have significant drawbacks: (1) They often require large amounts of annotated data for each specific action class, making them hard to generalize across different types of motions. However, unlike text-to-motion or human pose correction (which can be annotated through simple pipelines [31]), human motion sequences involve temporal changes. Annotating the differences between these temporal changes is challenging. (2) Many of these methods are limited to analyzing static poses or images rather than dynamic sequences of motion, reducing their applicability to real-world scenarios where movement dynamics are crucial.

LLMs, such as Llama [30], have shown potential in generating corrective instructions using few-shot or zero-shot learning. However, without proper fine-tuning and additional modalities, LLMs struggle to understand the spatial and temporal context of poses and motions, limiting their effectiveness in specialized fields like coaching or corrective instruction generation.

To address these limitations, we propose a novel approach, *CigTime*, for generating motion corrective instructions. Our method leverages existing motion editing pipelines to create datasets of motion triplets (source, target, and instruction). The key components of our approach include: **Motion-Editing-Based Data Collection**: We develop a pipeline that uses motion editing techniques to generate large datasets of motion pairs and corresponding corrective instructions. This process involves using a pre-trained motion editor to modify source motions according to generated instructions, resulting in target motions that reflect the desired corrections. **Fine-Tuning Large Language Models**: We fine-tune a large language model (LLM) on the generated datasets to enable it to produce precise and actionable corrective instructions. By training the LLM on a diverse set of motion sequences and corrections, we enhance its ability to understand and generate contextually relevant feedback.

In summary, our contributions include:

- We introduce a motion-editing-based pipeline to efficiently generate large datasets of corrective instructions, reducing the dependency on extensive manual annotations.

- We propose a general motion corrective instruction generation method which utilizes a large language model to translate motion discrepancies into precise and actionable instructional text, addressing the relationship between language and dynamic motions.

- Through comprehensive evaluations, we show that our method significantly outperforms existing models in generating high-quality corrective instructions, providing better guidance for users in various real-world scenarios.

## 2 Related work

### 2.1 Text Conditioned Human Motion Generation.

Conditional motion generation aims to synthesize diverse and realistic motion conditioning on different control signals, such as music [25, 26, 27, 28], action categories [2, 15, 21], physical signals [10, 16, 33]. Recent years have seen significant progress in text conditioned human motion generation [1, 2, 12, 13, 14, 20, 24, 34, 39, 40, 44, 45, 47]. Some methods [1, 12, 39] align the texts and motions into a joint embedding space for generation. Benefiting by aligning motion latent to the CLIP [35] embedding space, MotionCLIP [39] could generate out-of-distribution motions. Several works utilize other mechanisms to increase the diversity and quality of generated motions. TEMOS [34] and TEACH [2] employ transformer-based VAEs to generate motion sequences based on texts. Guo *et al.* [13] propose an auto-regressive conditional VAE to generate human motion sequences. Inspired by the achievements in image generation, the diffusion models, such as MotionDiffuse [45], MDM [40] and FLAME [24], have also been applied to motion generation. Some follow-up works [37, 43] attempt to improve the controllability of the diffusion model. Recently, the Vector Quantized Variational Autoencoder (VQ-VAE) has gained significant traction in being used to convert 3D human motion into motion tokens which are subsequently employed alongside language models. TM2T [14] proposes using these quantized tokens to facilitate the mapping between text and motion. T2M-GPT [44] employs an auto-regressive method to predict the next-index token. Further, MotionGPT [20, 47] utilizes large language models (LLMs) to simultaneously handle different motion-related

tasks. Recently, AvatarGPT [48] extends the generation models to unify high-level and low-level motion-related tasks, which supports human motions generation, prediction and understanding.

## 2.2 Motion Editing

Motion editing enables users to interactively refine generated motions to suit their expectations. PoseFix [7] utilize neural networks to edit 3D poses. Holden *et al.* [17] employs an autoencoder to optimize the trajectory constraints. MDM [40], MotionDiffuse [45] and FLAME [24] involve processing by masks that designate parts for editing through reverse diffusion. GMD [22] and PriorMDM [37] are designed to edit motion sequences conditioned on joint trajectories. OmniControl [43] incorporates control signals that encourage motions to conform to the spatial constraints while being realistic. Recently, FineMoGen [46] tackles fine-grained motion editing which allows for editing the motion of individual body parts, however its heavy reliance on specific-fine grained format limits the smooth coordination among movements of different body parts.

## 2.3 Corrective Instruction Generation

Traditional methods [6, 11] focus on specific action categories by building statistical models that require expert experience. These methods struggle to scale and generalize to various actions. Pose Tutor [8] uses neural networks to learn statistical models but requires large amounts of data for each action and can only analyze static images or poses. FixMyPose [23] creates a dataset with human-annotated corrective instructions on synthetic 2D images. PoseFix [7] designs an automatic annotation system and a conditioned auto-regressive model for corrective instruction generation, but it is limited to static poses. Recently, Large Language Models (LLMs) [30] have made significant advances in text generation. With appropriate prompting, LLMs can generate pose corrective instructions with few-shot or zero-shot example data. However, LLMs' access to text makes them less aware of a variety of possible motions that people could perform and links them with languages.

Our key insight for corrective instruction generation is to regard this task as a close yet inverse problem to text-conditioned motion generation and editing, allowing us to bring the progress in that fast-growing space to this understudied problem: We first propose a novel corrective instruction data collection pipeline based on motion editing. Subsequently, we design a model that leverage large language models to provide corrective instructions on spatial form and temporal dynamics.

## 3 Method

### 3.1 Overview

We present an overview of our approach in Fig. 1. Given a source motion sequence, $x^I \in \mathbb{R}^{T \times D}$, where $T$ is the number of frames and $D$ is the dimensionality of the motion representation, and a target motion sequence, $x^O \in \mathbb{R}^{T \times D}$, as input, our goal is to learn a function $\mathcal{T}$ which maps $x^I$ and $x^O$ to the corrective text instruction $L$, i.e., $\mathcal{T}(x^I, x^O) = L$.

To achieve this, we employ a pre-trained motion editor, which takes as input the source motion sequence and ground-truth corrective text, to output target motion sequences. Next, we quantize the source and target motion sequences into discrete tokens using a VQ-VAE-based network. Finally, we organize these tokens with a predefined template to fine-tune an LLM on the triplets that contain source motion sequence $x^I$, target motion sequence $X^O$, and corrective instruction $L$ for generating instructions that can efficiently modify the source to the target motion sequence.

### 3.2 Motion-Editing-Based Data Collection

The task of generating corrective instructions requires triplet data consisting of the source motion, the target motion, and the corrective instruction. Collecting such a dataset through human annotation is costly and inefficient. We aim to leverage existing pre-trained models to streamline the data collection process. However, there isn't an existing model that generates such triplets.

Our fundamental insight is to treat corrective instruction generation as an inverse process of motion editing, which uses a given text to guide an agent in editing its initial motion. We utilize the motion editing process to gather required triplets: we collect a set of source motions and employ a pre-trained

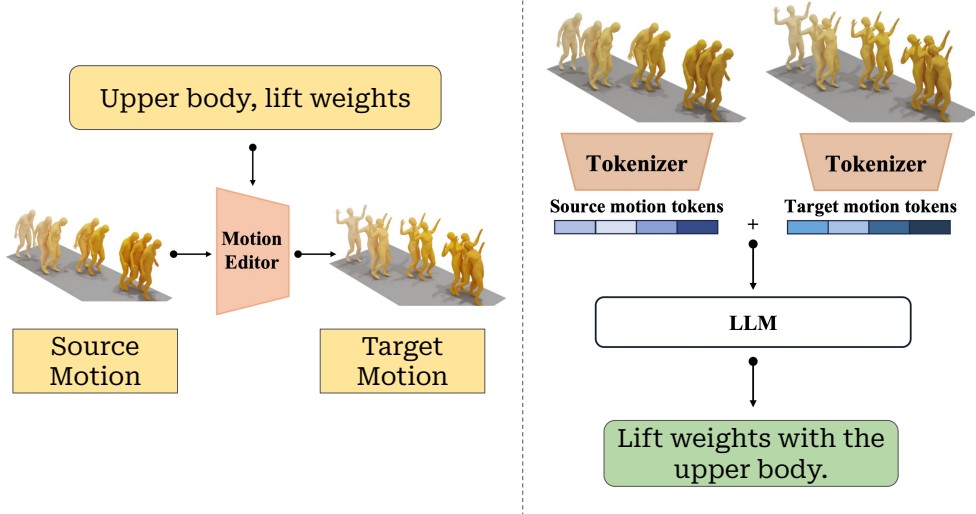

Figure 1: **Overview of CigTime**. Left: We leverage source motion tokens and corrective instructions as input to a motion editor to produce target motion tokens. Right: We then employ a language model to generate precise corrective instructions based on a given source and target motion. We demonstrate in the example generating corrective instructions for lifting weights with the upper body.

motion editor to edit the source motion based on a corrective instruction, resulting in the target motion.

**Motion Editing**   In this work, we utilize the motion diffusion model (MDM) [40] as the motion editor. Given the input motion sequence, $x$, and generation condition, $c$, MDM uses probabilistic diffusion models for motion generation. It comprises a forward process, which is a Markov chain involving the sequential addition of Gaussian noise to the data, and a reverse process that progressively denoises the data to get the edited motion. The forward process of MDM is formulated as,

$$q(x_{1:T}|x_0) = \prod_{t \geq 1} q(x_t|x_{t-1}), \tag{1}$$

$$q(x_t|x_{t-1}) = \mathcal{N}(\sqrt{\alpha_t}x_{t-1}, (1 - \alpha_t)\mathbf{I}), \tag{2}$$

where $\alpha_t \in (0, 1)$ are constant hyper-parameters. Further, the reverse process is formulated as,

$$p_\theta(x_{0:T}) = p(x_T) \prod_{t \geq 1} p_\theta(x_{t-1}|x_t), \tag{3}$$

$$p_\theta(x_{t-1}|x_t) = \mathcal{N}(x_{t-1}; \mu_\theta(x_t, t), \sigma_t^2\mathbf{I}), \tag{4}$$

where $\theta$ is the learnable parameters of the diffusion model, which gradually anneals the noise from a Gaussian distribution to the data distribution. We train MDM as a conditional generator $\mathcal{G}(x_t, c, t)$ that outputs $x_0$, where $c$ is the text condition, to maximize $p_\theta(x_{0:T})$.

In inference, MDM takes noise $n$ as $x_T$ and applies the reverse process to denoise the input based on the text condition, $c$, generating the motion sequences, $x_0$, corresponding to $c$. For the motion editing task, we utilize the corrective instruction, $L$, as the generation condition, $c$, to generate the corresponding corrective motion sequence, $x^L$. We then calculate the target motion sequence, $x^O$, by combining the source motion sequence, $x^I$, and the corrective motion sequence, $x^L$,

$$x^O = m \odot x^L + (1 - m) \odot x^I, \tag{5}$$

where $m$ is the joint mask for the body part $P$, and $\odot$ is the element-wise multiplication for masking operation. Through the above process, we are able to collect a large amount of $\langle x^I, x^O, L \rangle$ triplets. We use this dataset to fine-tune a large language model (LLM) for the corrective instruction generation task as in the following.

Below is an instruction that describes a task, paired with an input that provides further context. Write a response that appropriately completes the request.

### Instruction:
I will give you two motion sequences, representing sequences of the same character doing different actions. You are asked to compare two sequences and output what modifications the person should make to transfer from the first action to the second action.

### Input:
Action 1: [Token list for source motion sequence]
Action 2: [Token list for Target motion sequence]

### Response:
[Correctional Instruction]

Figure 2: **Template for LLM fine-tuning.** The LLM is required to output the corrective instructions given token lists for the source and target motion sequences (i.e., Action 1 and Action 2) as well as instructions on the expected output.

## 3.3 Fine-tuning LLMs for Corrective Instruction Generation

With the prepared dataset of triplets from the motion editing process, we learn the inverse process of motion editing, a function, $\mathcal{T}$, that maps source and target motion sequence pairs to corrective instructions. We first learn an encoder based on VQ-VAE [41] to tokenize the motion sequences into discrete tokens and organize the discrete tokens based on a pre-defined template. Then, we fine-tuned an LLM to generate the corrective instruction, $L$, based on the tokens of the source and target motion sequence, $x^I$ and $x^O$.

**Tokenizer Pre-training** Compared to directly feeding the original data to the LLMs, the discrete representation has been proven to be more suitable for fine-tuning LLMs with human-motion-related tasks [20, 47]. Inspired by these works, we initialize a VQ-VAE-based network, which contains an encoder $\mathcal{E}$, a codebook $C$, and a decoder $\mathcal{D}$. The encoder $\mathcal{E}$ takes motion sequence, $x$, as input and maps, $x$, into discrete features, $f \in \mathcal{T} \times \mathcal{H}$, where $H$ is the dimensionality of the frame feature.

The codebook, $C \in \mathbb{R}^{K \times H}$, represents different codes, where $K$ is a predefined number of different discrete codes and $c_k \in \mathbb{R}^H$ is the k-th code. VQ-VAE quantizes the discontinuous feature $f$ to the discrete latent codes, $z \in \mathbb{R}^{T \times H}$, through codebook, $c$, by projecting each per-frame feature $f_i$ to its nearest code:

$$z_i = Q\left(f_i\right) = c_k, \text{ where } k = \operatorname{argmin}_j \|f_i - c_j\|_2^2, \tag{6}$$

where $Q$ represents the quantization operation. The decoder $\mathcal{D}$ takes the code, $z$, as input, and reconstructs the motion sequence, $x^{O'}$. We use the index $k$ as the token of each discrete code, $c_k$, as the token representation of the frame feature $f_i$. We apply the L2 loss for the training of the tokenizer,

$$\mathcal{L}^{recon} = ||x^O - x^{O'}||_2^2. \tag{7}$$

Considering that the quantization operation disrupts gradient backpropagation, we employ an exponential moving average (EMA) [42] for the codebook update and stabilize the training process. Besides, we apply the commitment loss [41] to update the tokenizer encoder,

$$\mathcal{L}^{com} = \sum_{i=1}^{T} \|f_i - sg(z_i)\|_2^2, \tag{8}$$

where $sg(\cdot)$ is the stop gradient operation that helps stabilize the training process.

**Fine-tuning LLM** Instruction Tuning is a widely used technique to enable LLMs to handle specific tasks. In this work, we employ this technique to fine-tune our LLM. Specifically, given an LLM, $\mathcal{T}$, a source discrete token set, $I^s = I_0^s, I_1^s, ..., I_{n^s}^s$, and a target discrete token set, $I^t = I_0^t, I_1^t, ..., I_{n^t}^t$, we

organize the input of $\mathcal{T}$ to follow the template as shown in Fig. 2. This input is then tokenized into text tokens $U^I = u_0^I, u_1^I, ..., u_{n^{U^I}}^I$. Additionally, we tokenize the ground-truth corrective instruction, $L$, into text tokens, $U^O = u_0^O, u_1^O, ..., u_{n^{U^O}}^O$.

The LLM processes the input tokens, $U^I$, and auto-regressively predicts the probability distribution of the next tokens $p_{\mathcal{L}}(u|U^I) = \prod_j p_{\mathcal{L}}(u_j^O|u_{0:j-1}^O, U^I)$. During training, we maximize the log-likelihood of the data distribution by applying cross-entropy loss:

$$\mathcal{L}^{LLM} = -\sum_{j=0}^{U^O} \log p_{\mathcal{L}}(u_j^O|u_{0:j-1}^O, U^I). \tag{9}$$

By using a structured input template and optimizing the cross-entropy loss, we enable the LLM to generate accurate and contextually relevant corrective instructions. This approach ensures that the model effectively learns to convert discrepancies between the source and target motions into precise and actionable instructional text.

**Learning Representation for Motion Tokens**   Previous methods for training text-to-motion models involve either using an existing vocabulary for motion tokens [47] or assigning new learnable embeddings [20, 48], followed by fine-tuning with techniques like LoRA. We tried both approaches, but the results of utilizing one of them alone were not satisfying. There are two main reasons: First, using a fixed vocabulary and embeddings prevents capturing the correlation of motion differences and corrective instructions, as the weights are trained on tasks with a large domain gap. Second, while new embeddings can be learned with LoRA, the distribution of the original vocabulary's embeddings imposes constraints, making the learned embeddings suboptimal, especially given the smaller scale of training data for corrective instructions.

To address these challenges, we integrate the goods of both. We use existing vocabulary tokens for their rich semantics and fine-tune all embeddings to maximize performance and reduce the domain gap. We also introduce an anchor loss to prevent the embeddings from diverging:

$$\mathcal{L}^{Anchor} = \lambda \cdot \|W - W_0\|_2^2, \tag{10}$$

where $\lambda$ is a regularization coefficient that controls the influence of loss, $W_0$ represents the network weights before training, $W$ represents the network weights after training.

## 4   Evaluation

### 4.1   Experiment Setup

**Datasets** We obtain the source motion sequences from HumanML3D [13], a dataset containing 3D human motions and associated language descriptions. We make use of the entire dataset for the collection of source motions. We then generate triplets based on pre-trained motion editor with instructions and target motions. We split HumanML3D following the original setting and for each motion sequence in HumanML3D, we randomly select one instruction from the corresponding split for editing the sequence. We subsequently edit the source motion sequences with MDM [40] conditioned on the corrective instructions to obtain the target sequences.

**Implementation Details** We fine-tune a pre-trained Llama-3-8B [30] using full-parameter fine-tuning for corrective instruction generation. The model is optimized using the Adam optimizer with an initial learning rate of $10^{-5}$. We use a batch size of 512 and train on four NVIDIA Tesla A100 GPUs for eight epochs, which takes approximately 5 hours to complete. Following HumanML3D [13], the dimensionality, $D$, of the motion sequences is set to 263 for our experiments.

**Evaluation Metrics** We evaluate the generated corrective instruction with two types of metrics.

(1) Corrective instruction quality: BLEU [32], ROUGE [29], and METEOR [4] are commonly employed metrics that assess various n-gram overlaps between the ground-truth text and the generated text. Although these metrics focus on structural text similarity, they tend to disregard semantic meaning. Consequently, we also utilize the cosine similarity of text CLIP embeddings as an evaluation metric to better compare semantic similarity.

(2) Reconstruction accuracy: To evaluate the quality, we use the generated corrective instruction as an editing condition to modify the source motion sequences and obtain the generated target motion. We then compare this with the ground-truth target motion. Specifically, we employ Mean Per Joint Position Error (MPJPE) to measure the average Euclidean distance between the generated and ground-truth 3D joint positions for all joints. Additionally, we calculate the Fréchet Inception Distance (FID) using a feature extractor [13] to evaluate the distance between the feature distributions of the generated and ground-truth target motions. Ideally, the generated motion sequences should closely resemble the target motion sequences.

Table 1: **Comparison to the Existing Work.** We compare our approach against large language (Llama-3-8B, Llama-3-8B-LoRA, Qwen-7B, Mistral-7B) and motion-language (MotionGPT, MotionGPT-M2T) models. We demonstrate that our approach, *CigTime* outperforms all the baselines by a large margin for corrective instruction generation for human motion.

| Method | Instruction Quality | | | | Reconstruction Accuracy | |
|---|---|---|---|---|---|---|
| | BLEU ↑ | ROUGE↑ | METERO ↑ | CLIPScore ↑ | MPJPE ↓ | FID ↓ |
| Ground-Truth | 1.00 | 1.00 | 1.00 | 1.00 | 0.00 | 0.00 |
| Llama-3-8B | 0.15 | 0.29 | 0.45 | 0.77 | 0.21 | 3.04 |
| Llama-3-8B-LoRA | 0.10 | 0.19 | 0.36 | 0.77 | 0.24 | 2.09 |
| Mistral-7B | 0.16 | 0.30 | 0.46 | 0.80 | 0.22 | 5.03 |
| Mistral-7B-LoRA | 0.08 | 0.19 | 0.27 | 0.79 | 0.75 | 1.84 |
| MotionGPT | 0.02 | 0.10 | 0.11 | 0.76 | 0.80 | 8.84 |
| MotionGPT-M2T | 0.02 | 0.13 | 0.12 | 0.76 | 1.05 | 7.96 |
| Ours | **0.24** | **0.35** | **0.52** | **0.82** | **0.13** | **1.44** |

**Comparison Baselines** To the best of our knowledge, we are the first to generate corrective instruction for general motion pairs. Thus, we adopt two different kinds of methods designed for general text-based tasks and motion captioning.

(1) Llama3 [30], Qwen [3] and Mistral [19] are all large language models designed for general text-based tasks. They can be applied to unseen tasks with just a few-shot data. We utilize the in-context learning technique [9] to generate correction instructions by giving them examples of the source-target-instruction triplets. We present the detailed prompts in the supplemental material. In addition to the baselines that use in-context learning with LLMs, we ablate different fine-tuning techniques. To do so, we compare our approach, which uses full-parameter LLM tuning to a variant, which utilizes the LoRA adapter [18] to fine-tune the Llama 3 8B and Mistral 7B models.

(2) MotionGPT [20]. Although MotionGPT isn't trained with corrective instruction data, it has been proven to have the ability to generalize across different motion-based tasks by utilizing specific input templates for different tasks. Thus, we adopt this method for corrective instruction generation by utilizing the template mentioned in Section. 3.3. In addition, as generating corrective instructions is not a target for MotionGPT, we create yet another baseline called MotionGPT-M2T that employs MotionGPT to generate captions corresponding for the target motions.

## 4.2   Quantitative Results

Our quantitative results are presented in Table. 1. We further discuss below the quality of the corrective instructions and the reconstruction accuracy of target motion after editing.

**Corrective Instruction Quality**   Our method demonstrated superior performance across most metrics when compared to baseline methods, as presented in Table. 1. Specifically, our method achieved the highest BLEU-4, ROUGE-2 and METERO scores of 0.24, 0.35, and 0.52, significantly surpassing the baseline methods. This indicates that our method generates text with higher precision.

Furthermore, our method achieved the highest CLIP Score of 0.82, outperforming other baselines. The CLIP Score indicates the semantic alignment of the generated text with visual content, and a higher score demonstrates better performance in maintaining this alignment.

We find that the two baselines adopted from MotionGPT both present inferior performances, which can be attributed to its training on a text-motion dataset, which lacks the capability to compare

two motion sequences and identify specific differences. Besides, although MotionGPT excels at generating captions for motion sequences, it's still difficult to reconstruct the original target motion sequence from the generated descriptions. This is because describing the differences and similarities between two motion sequences can help us accurately depict the target motion with fewer statements, which MotionGPT does not possess.

This evidenced that simply fine-tuning Llama-3 using the generated data would not result in a satisfactory corrective instruction generation, e.g., due to overfitting or catastrophic forgetting. Although the outputs can induce similar target motion sequences compared to the ground truth, the increased variance in the text can lead to a decrease in the overall NLP metrics such as BLEU, ROUGE, and METERO.

Overall, these results highlight the effectiveness of our method in generating high-quality corrective instructions, with significant improvements in precision, similarity, and visual-semantic consistency over the baseline methods.

Table 2: **Ablation study with different network structure**. We extend the LLMs' vocabularies with new learnable embeddings for the motion tokens and update the corresponding embeddings during fine-tuning as baselines. We also compare variants that utilizes T5 as the backbone (ours-T5), and continous representaion (Ours-Continuous).

| Method | Instruction Quality | | | | Reconstruction Accuracy | |
|---|---|---|---|---|---|---|
| | BLEU ↑ | ROUGE↑ | METERO ↑ | CLIPScore ↑ | MPJPE ↓ | FID ↓ |
| Llama-3-8B-Extended | 0.12 | 0.23 | 0.44 | 0.80 | 0.27 | 5.43 |
| Mistral-7B-Extended | 0.18 | 0.27 | 0.42 | 0.81 | 0.19 | 1.45 |
| Ours-Extended | **0.24** | **0.37** | **0.55** | **0.84** | 0.16 | 1.50 |
| Ours-Continuous | 0.12 | 0.24 | 0.47 | 0.78 | 0.20 | 2.56 |
| Ours-T5 | 0.14 | 0.25 | 0.46 | 0.80 | 0.33 | 5.03 |
| Ours | **0.24** | 0.35 | 0.52 | 0.82 | **0.13** | **1.44** |

**Reconstruction Accuracy**  The evaluation of reconstruction accuracy highlights the superior performance of our method in distinguishing between source and target motions. As shown in Table 1, our method achieved the lowest MPJPE of 0.1330, indicating the highest accuracy in pose reconstruction. Furthermore, our method also attained the lowest FID - Target score of 1.4442, demonstrating its effectiveness in generating data that closely matches the target motion. Similarly, MotionGPT's inferior performance in these metrics is a result of its limited ability to analyze differences between motion pairs, as evidenced by its MPJPE of 0.8011 and FID score of 8.8350.

Additionally, although LLM models like Llama-3-8B can maintain text consistency via in-context learning, they are unable to grasp the intricate connections between motion sequences and language, leading to inferior overall performance compared to our approach. Even when benefiting from fine-tuning through LoRA, these models still cannot generate high-quality corrective instructions.

Overall, these results underline the effectiveness of our method in accurately distinguishing and reconstructing the differences between source and target motions, outperforming the baseline methods in both MPJPE and FID metrics.

Table 3: **Ablation study with different motion editors.** We assess the reconstruction accuracy of various methods employing different motion editors for evaluation.

| Method | MDM | | PriorMDM – LW | | PriorMDM – RF | |
|---|---|---|---|---|---|---|
| | MPJPE ↓ | FID ↓ | MPJPE ↓ | FID ↓ | MPJPE ↓ | FID ↓ |
| Ground-Truth | 0.00 | 0.00 | 0.22 | 2.97 | 0.25 | 5.22 |
| Llama-3-8B-LoRA | 0.24 | 2.09 | 0.27 | 3.08 | 0.37 | 7.08 |
| MotionGPT | 0.80 | 8.84 | 0.80 | 9.80 | 0.77 | 19.07 |
| MotionGPT-M2T | 1.05 | 7.96 | 0.80 | 8.48 | 0.74 | 28.95 |
| Ours | **0.13** | **1.44** | **0.22** | **3.02** | **0.26** | **5.34** |

**Ablation study with different network structurer**  To validate that our token embedding training method is superior to the extended token embedding approach used in previous algorithms, we

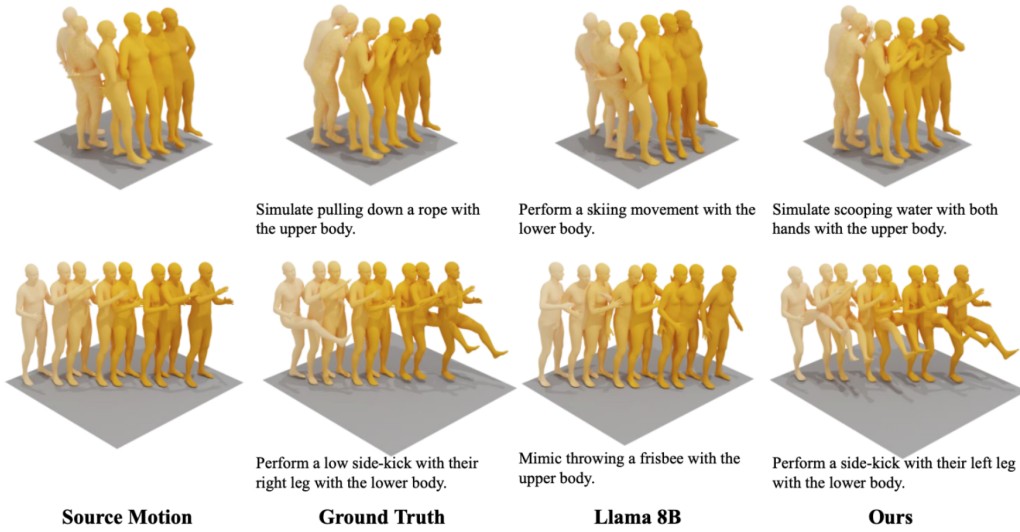

Figure 3: Visualization of corrective instructions and reconstructed motions for different methods.

conducted a comparison of LLMs trained using token embeddings, as shown in Table 2. Although fine-tuning with extended vocabulary can enhance the text-based metrics, these instructions cause a decline in the motion editing performance, resulting in a reduction in MPJPE and FID. From the perspective of the task definition, we require a model that prioritizes high reconstruction quality over instruction quality. Therefore, extending vocabulary is more detrimental than beneficial for our task.

Besides we fine-tune T5-770M [36], as in Motion-GPT [20] and AvatarGPT [48] to validate the impact of different LLM frameworks on the results. The experimental results show that the T5 framework does not offer an advantage over larger language models [30] in the Motion Corrective Instruction Generation task. We also compared our method with its variant based on continuous representations, as implemented by MotionLlm [5]. As observed, our method still outperforms the continuous baseline across all the reconstruction accuracy metrics.

**Evaluation with Different Motion Editors**   Different people may perform various actions in response to the same instruction. Our goal is for our model to produce instructions that are as accurate and widely accepted as possible. Therefore, we evaluate our methods and baselines using different motion editors. In addition to MDM, which we used to generate the ground-truth dataset, we also assess the methods with two different versions of PriorMDM as shown in Table 3.

Our proposed method consistently outperforms other models across different motion editors, demonstrating the lowest MPJPE and FID values, close to the ground truth. This highlights its effectiveness in generating accurate and visually similar corrective motions. In contrast, models like MotionGPT and its variant exhibit significantly higher errors, indicating limitations in their generation capabilities.

### 4.3   Visual Results

To further analyze the performance of different methods, we present visual comparisons in 3. As shown in the results, our algorithm largely maintains similar semantics and achieves reconstruction results that are closely aligned with the ground truth. This demonstrates the accuracy of our algorithm in generating corrective instructions. In contrast, Llama3-8B, despite achieving favorable numerical results, may incorrectly identify the joint parts involved in motion editing. This highlights our approach's superiority in providing accurate and contextually appropriate motion corrections.

## 5   Conclusion

We introduced a new task and a framework for generating corrective instructions that translate a source motion into a target motion. Our key insight is to leverage the fast growing field of text-conditioned

motion-editing for this related yet understudied inverse problem. To create a dataset for this task, we proposed a motion editing pipeline that minimizes the need for extensive manual annotations. We demonstrated the utility of our approach which largely outperforms existing related models.

While our work provides a strong foundation for corrective instruction generation in human motion, there are limitations to our framework.

1. First, the curated dataset captures differences between source and target motions, but it lacks targeted feedback on form and dynamics that are specific to actions and sports, which require more detailed and subtle instructions for learning particular skills.

2. Second, due to the limitation of the pretrained motion editor [40], we can only handle source and target motion pairs with the same sequence length, without context or scenes.

3. Third, the corrective instruction generation method may be misused to generate instructions for insulting or inappropriate motions.

We aim to address these limitations in future research, along with further advances of text-conditioned motion-editing frameworks, which share our challenges, limitations, and potential solutions.

**Acknowledgment**

This work is supported by the Early Career Scheme of the Research Grants Council (grant # 27207224), the HKU-100 Award, a donation from the Musketeers Foundation, an Academic Gift from Meta, and the Microsoft Accelerate Foundation Models Research Program. The authors would like to thank Robert Wang, Shugao Ma, Alexander Winkler, Yijing Li, and Chris Twigg for their valuable discussions. Special thanks are also extended to Pei Zhou, Chenming Zhu, and Shumin Sun for their support in the real-world application.

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

Figure 4: **In-context learning for corrective instruction generation.** The prompt for the LLMs in in-context learning includes a task description and several examples. This information is given to the LLMs, instructing them to generate correctional instructions for new motion pairs.

## A  Prompt for the LLMs In-context Learning

To enable large language models (LLMs) for generating correctional instructions grounded in given sequences, we apply the in-context learning technique [9]. This method allows LLMs to make predictions based on contexts supplemented with a limited number of examples. The prompt used for in-context learning is displayed in Figure 4.

## B  Additional Experiments

### B.1  Generalization to New Data

Our algorithm is fully trained and tested on the HumanML3D [13] dataset, which may impact its generalization. To evaluate the generalization ability of our algorithm, we collected 1525 samples from the Fit3D [11] dataset.

We present the results in Table 4. These results show that the BLEU, ROUGE, and METEOR scores decreased from 0.24, 0.35, and 0.52 to 0.03, 0.05, and 0.20, respectively. This indicates that when the dataset changes, the corrective instructions generated by our algorithm deviate from the ground truth in form. However, the changes in CLIP score, MPJPE, and FID are subtle. This suggests that even after switching datasets, our algorithm can still effectively capture the differences in motion pairs and describe them in appropriate language. Our algorithm therefore generally showcases a notable level of generalization capability.

### B.2  Experimental Results on KIT Dataset

We further evaluate our method baselines on KIT dataset. As shown in Table 5 , our method still outperforms other baselines across all metrics, demonstrating the generalization capability.

Table 4: Numeric Results

| Method | Dataset | Instruction Quality | | | | Reconstruction Accuracy | |
|--------|---------|------|-------|--------|----------|---------|-------|
| | | BLEU ↑ | ROUGE↑ | METERO ↑ | ClipScore ↑ | MPJPE ↓ | FID ↓ |
| Ours | Humanml3D | 0.24 | 0.35 | 0.52 | 0.82 | 0.13 | 1.44 |
| Ours | Fit3d | 0.03 | 0.05 | 0.20 | 0.81 | 0.18 | 1.24 |

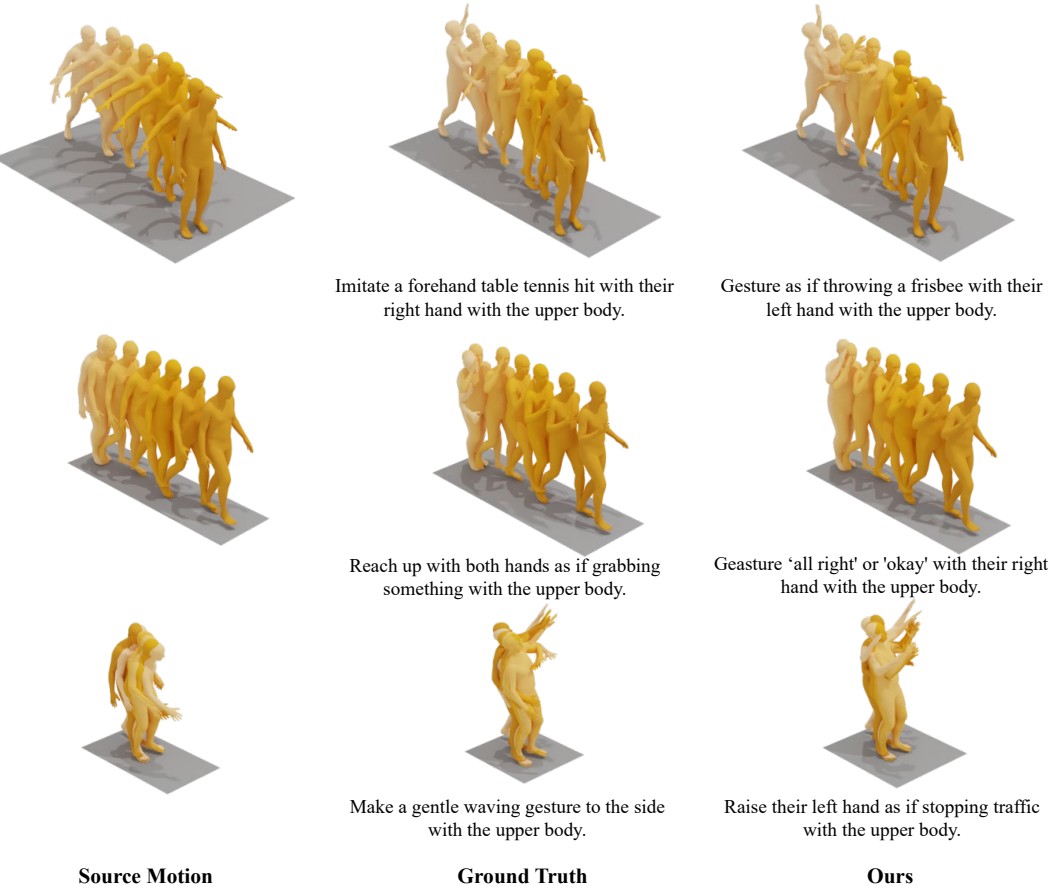

Imitate a forehand table tennis hit with their right hand with the upper body.

Gesture as if throwing a frisbee with their left hand with the upper body.

Reach up with both hands as if grabbing something with the upper body.

Geasture 'all right' or 'okay' with their right hand with the upper body.

Make a gentle waving gesture to the side with the upper body.

Raise their left hand as if stopping traffic with the upper body.

**Source Motion**                **Ground Truth**                **Ours**

Figure 5: **Diversity of the corrective instructions.** We present some examples where the reconstructed motions have a similar appearance to the target motions, but the corrective instructions still differ from the ground truth, demonstrating the robustness of our approach generating effective and semantically meaningful corrective instructions.

Table 5: **Experimental results on KIT dataset**. We conduct a comparative analysis of our method against baselines on the KIT dataset.

| Method | Instruction Quality | | | | Reconstruction Accuracy | |
|---|---|---|---|---|---|---|
| | BLEU ↑ | ROUGE↑ | METERO ↑ | CLIPScore ↑ | MPJPE ↓ | FID ↓ |
| Llama-3-8B-LoRA | 0.11 | 0.17 | 0.36 | 0.78 | 0.37 | 5.03 |
| Qwen-1.5-7B-LoRA | **0.14** | 0.25 | 0.46 | **0.80** | 0.33 | 5.03 |
| Mistral-7B-LoRA | 0.13 | 0.17 | 0.36 | 0.79 | 0.30 | 5.02 |
| Ours | **0.14** | **0.27** | **0.47** | **0.80** | **0.21** | **4.52** |

Figure 6: **Real-world application**. This figure illustrates the source and target motions collected from real-world participants, alongside the corrective instructions generated by different methods. Left to right: the source motion, target motion, generated corrective instruction, and the corrected motions. We collect the videos with a single camera and extract motions with WHAM.

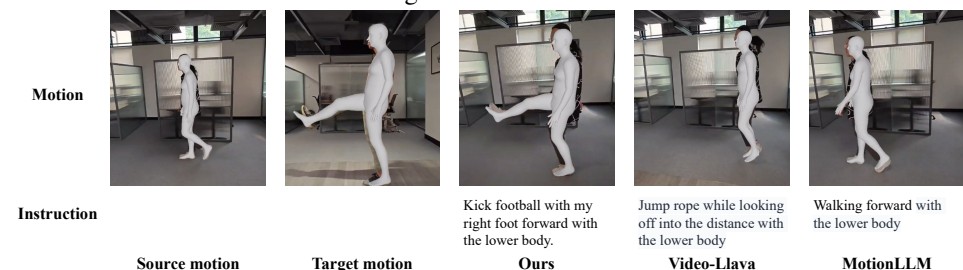

B.3 Additional Visual Results

We present visualization examples of our corrective instructions and reconstructed motion sequences in Fig. 5. We observe that although the corrective instructions predicted by our algorithm sometimes differ from the ground truth (e.g., "forehand table tennis" versus "throwing a frisbee"), they can still result in remarkably similar modified motions. In specific frames, the resulting motions are nearly identical, as seen in the beginning and ending frames of the first example. This phenomenon aligns with real-world scenarios where individuals can provide multiple, semantically distinct suggestions that lead to similar corrective outcomes when correcting others' mistakes. This underscores the robustness of our approach in generating effective motion corrections, even when the specific instructions vary.

Considering the diversity of correction instructions, traditional metrics such as BLEU, ROUGE, or METEOR alone may not be sufficient to describe their correctness. Thus, we incorporated CLIP score and reconstruction metrics as supplementary evaluation measures, creating a more exhaustive benchmark for evaluating correction instruction generation. We present more visual results in Fig. 7 and 8.

## C Implementation Details

### C.1 Motion Editing

We utilize the pre-trained MDM [40] for motion editing. The model is trained on the HumanML3D dataset with a batch size 32, a learning rate 0.0002, and training for 50000 epochs. In the editing process, we set the maximum reverse diffusion steps to 50.

### C.2 Architecture of Our Tokenizer

We utilize TCN-based structures for both encoder and decoders, which extract spatiotemporal features for human motion through convolution with a kernel size of 1. We also extract temporal features through dilation convolution and larger kernels (9 or 3). We list the details of our network architecture in Tab. 7. The decoders $\mathcal{U}$ and $\mathcal{B}$ share the same architecture.

## D Example of Corrective Instructions

We present some corrective instructions in Tab. 6.

Table 6: Examples of corrective instructions.

| Body part | Corrective instruction |
|---|---|
| Upper body | Gesture as if explaining a large concept with both hands. |
| | Act as if using a whip with their right hand |
| | Feign holding and adjusting a large telescope |
| | Performs a chest-expanding exercise, pulling arms back |
| | Fake a tennis serve |
| | Reenact painting a wall with a roller |
| | Act out swinging a cricket bat |
| | Shoot a bow and arrow |
| Lower body | Wade through water |
| | Simulate hopping over a turning jump rope |
| | Stand on their right leg briefly |
| | Step side to side, simulating dancing |
| | Act out getting on a bicycle |
| | Jump over a puddle |
| | Kick gently with the right foot |
| | Walk like a model on a runway |

# E  Real-world Application

Obtaining precise motion in real life is difficult. However, we find that existing motion estimation algorithms enable us to obtain usable motion sequences in most cases. To verify whether the current pipeline can be applied to real life, we conduct the following experiment.

We invited two participants, one acting as a coach and the other as a trainee. The trainee first performed a source motion sequence. Then, the coach was tasked with generating a target motion sequence that differed from the source sequence. We utilized a pose estimation algorithm (WHAM[38]) to extract these motion sequences and use our method to generate corrective instructions. The trainee is then required to correct his motion based on the corrective instructions. We present an example in Figure 6 of the global response pdf. In this example, it is evident that existing motion estimation algorithms can accurately estimate the motions of both the trainee and the coach. Furthermore, our algorithm is capable of understanding these motion sequences to provide appropriate corrective instructions.

Table 7: Architecture of the tokenizer.

| Components | Architecture |
| --- | --- |
| Linear Encoder | (0): Conv1D(J*3, 256, kernel_size=(3,), stride=(1,), padding=(1,))
(1): ReLU()
(2): 2 × Sequential(
  (0): Conv1d(256, 256, kernel_size=(3,), stride=(1,), padding=(1,))
  (1): ResConv1DBlock(
    (0): (activation1): ReLU()
    (1): (conv1): Conv1D(256, 256, kernel_size=(3,), stride=(1,), padding=(9,), dilation=(9,))
    (2): (activation2): ReLU()
    (3): (conv2): Conv1D(256, 256, kernel_size=(1,), stride=(1,)))
  (2): ResConv1DBlock(
    (0): (activation1): ReLU()
    (1): (conv1): Conv1D(256, 256, kernel_size=(3,), stride=(1,), padding=(3,), dilation=(3,))
    (2): (activation2): ReLU()
    (3): (conv2): Conv1D(256, 256, kernel_size=(1,), stride=(1,)))
  (3): ResConv1DBlock(
    (0): (activation1): ReLU()
    (1): (conv1): Conv1D(256, 256, kernel_size=(3,), stride=(1,), padding=(1,))
    (2): (activation2): ReLU()
    (3): (conv2): Conv1D(256, 256, kernel_size=(1,), stride=(1,)))) |
| Residual VQ | (0): (conv1): Conv1D(256, 16, kernel_size=(1,), stride(1,))
(1): (codebook_class): nn.Parameter((64, 16), requires_grad=False)
(2): (codebook_residual): nn.Parameter((64, 16), requires_grad=False)
(3): (conv2): Conv1d: Conv1D(16, 256, kernel_size=(1,), stride=(1,)) |
| Decoder | (0): 2 × Sequential(
  (0): Conv1d(256, 256, kernel_size=(3,), stride=(1,), padding=(1,))
  (1): ResConv1DBlock(
    (0): (activation1): ReLU()
    (1): (conv1): Conv1D(256, 256, kernel_size=(3,), stride=(1,), padding=(9,), dilation=(9,))
    (2): (activation2): ReLU()
    (3): (conv2): Conv1D(256, 256, kernel_size=(1,), stride=(1,)))
  (2): ResConv1DBlock(
    (0): (activation1): ReLU()
    (1): (conv1): Conv1D(256, 256, kernel_size=(3,), stride=(1,), padding=(3,), dilation=(3,))
    (2): (activation2): ReLU()
    (3): (conv2): Conv1D(256, 256, kernel_size=(1,), stride=(1,)))
  (3): ResConv1DBlock(
    (0): (activation1): ReLU()
    (1): (conv1): Conv1D(256, 256, kernel_size=(3,), stride=(1,), padding=(1,))
    (2): (activation2): ReLU()
    (3): (conv2): Conv1D(256, 256, kernel_size=(1,), stride=(1,))))
  (2) Conv1D(256, 256, kerne_size=(1,), stride=(1,))
(1): ReLU()
(2): Conv1D(256, 75, kernel_size=(1,), stride=(1,)) |

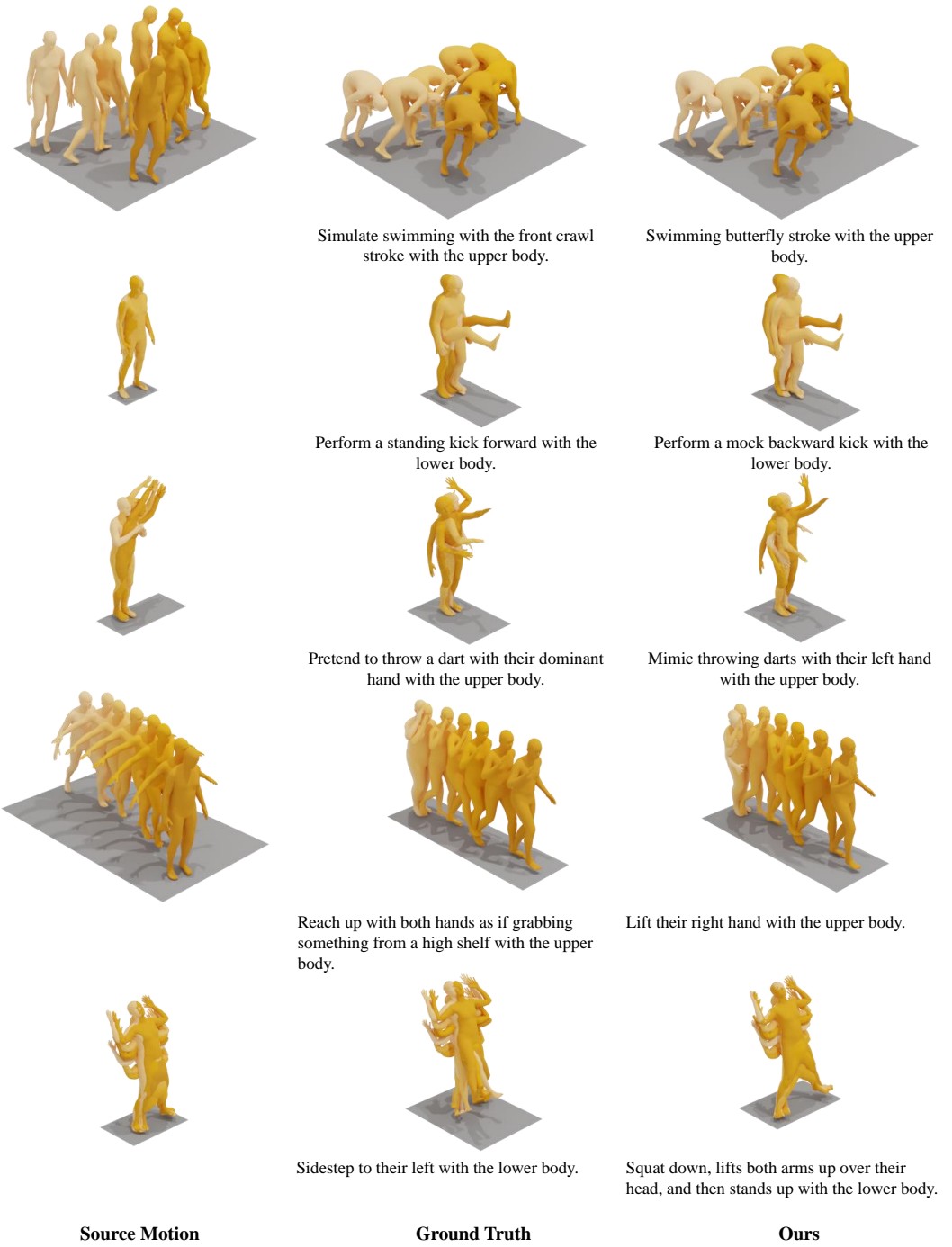

Simulate swimming with the front crawl stroke with the upper body.

Swimming butterfly stroke with the upper body.

Perform a standing kick forward with the lower body.

Perform a mock backward kick with the lower body.

Pretend to throw a dart with their dominant hand with the upper body.

Mimic throwing darts with their left hand with the upper body.

Reach up with both hands as if grabbing something from a high shelf with the upper body.

Lift their right hand with the upper body.

Sidestep to their left with the lower body.

Squat down, lifts both arms up over their head, and then stands up with the lower body.

**Source Motion**    **Ground Truth**    **Ours**

Figure 7: **Additional visualizations.** Qualitative results for the corrective instructions and reconstructed motion sequences.

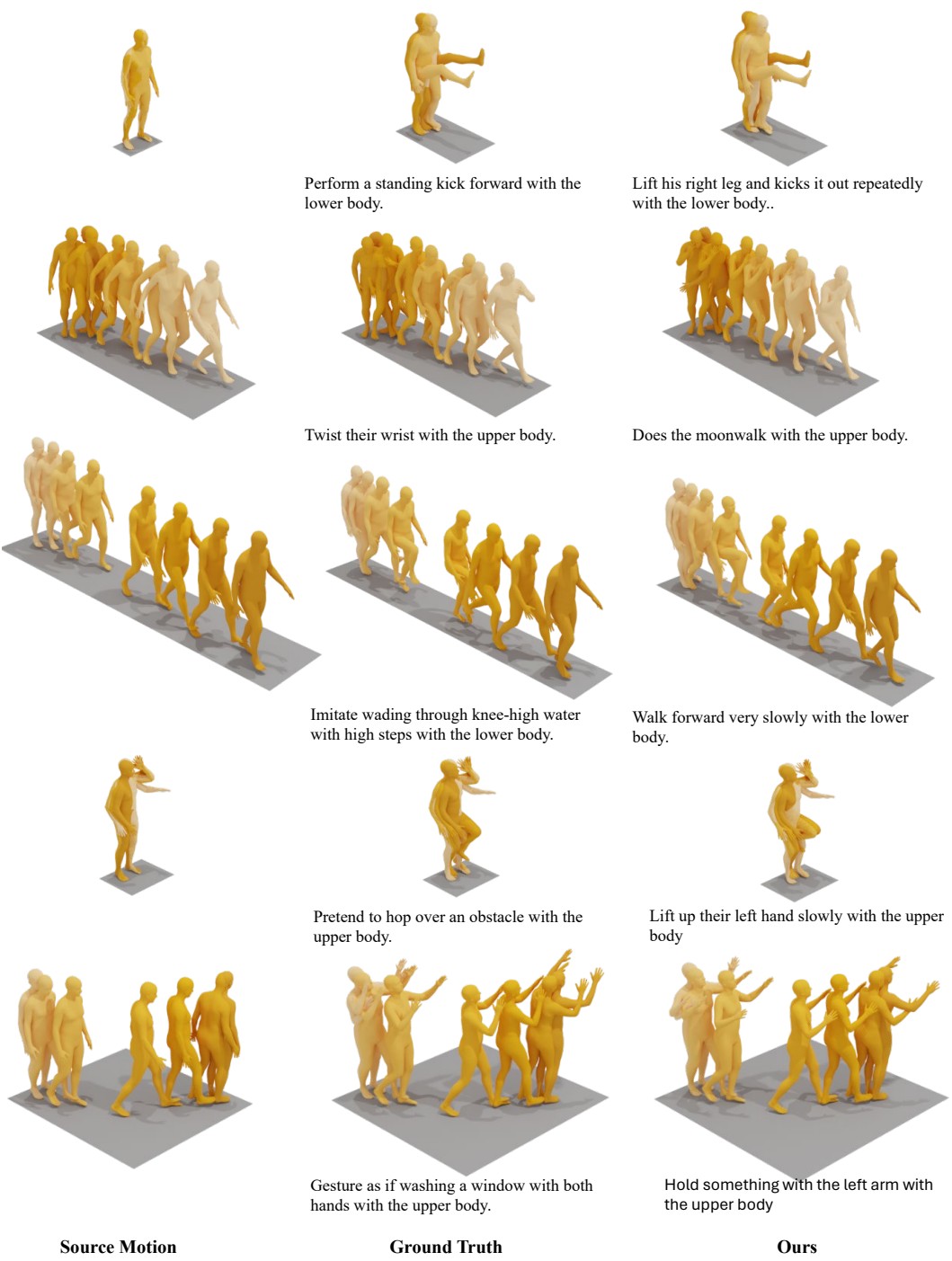

Perform a standing kick forward with the lower body.

Lift his right leg and kicks it out repeatedly with the lower body..

Twist their wrist with the upper body.

Does the moonwalk with the upper body.

Imitate wading through knee-high water with high steps with the lower body.

Walk forward very slowly with the lower body.

Pretend to hop over an obstacle with the upper body.

Lift up their left hand slowly with the upper body

Gesture as if washing a window with both hands with the upper body.

Hold something with the left arm with the upper body

**Source Motion**        **Ground Truth**        **Ours**

Figure 8: **Additional visualizations.** Qualitative results for the corrective instructions and reconstructed motion sequences.

