# OpenReview forum: "CigTime: Corrective Instruction Generation Through Inverse Motion Editing"
_NeurIPS.cc/2024/Conference — NeurIPS 2024 poster_

### Official Review · Reviewer_bYSe · 2024-07-11

**Soundness:** 3
**Presentation:** 2
**Contribution:** 2
**Rating:** 6
**Confidence:** 5

**Summary:**

The paper introduces a novel approach to generating corrective instructional text based on motion editing and generation models. This work is particularly motivated by the applications in sports coaching and motor skill learning. The authors propose a method that takes a user's current motion (source) and the desired motion (target) and generates text instructions to guide the user towards achieving the target motion. The approach leverages existing motion generation frameworks to compile datasets of triplets (source motion, target motion, and corrective text) and employs a large language model (LLM) fine-tuned on this data to produce corrective instructions.

**Strengths:**

1. This paper properly uses LLM and focuses on an important and novel application.

2. The author formulated this problem as an inverse problem of motion editing and used the pretrained editing model to create a data model.

**Weaknesses:**

1. The inputs of this work are motions, which are hard to obtain in real life. The off-the-shelf motion estimation algorithm may cause estimation errors, which limits the application range and fields.

2. The data-generating pipeline heavily depends on the motion-editing model, which may lead to error accumulating and overfitting.

**Questions:**

1. Could the authors try to compare with vision LLMs, like GPT4O, InternVL [1], and MotionLLM [2].

2. The way of using text pure LLM for comparison is in-context learning, which is not fair. How does an LLM understand the confused motion index tokens without fine-tuning?

3. The method and dataset heavily depend on the generalizable abilities of the motion editing model. How does the author make sure that the pretrained motion editing model can take flexible instructions generated by Chatgpt as input and achieve satisfactory editing results?

4. The authors take discrete token indexes as special tokens. This is hard to capture the detailed spatial and temporal information. Did the authors try to use continuous representations? Just like in LLaVA [3] and MotionLLM [2].

5. The authors use a pretrained motion editing model to generate data, then finetune an LLM using the data, and finally evaluate using the same motion editing model. The data-generating pipeline and evaluation use the same motion editing model. This may lead to heavy overfitting, which can be seen in Figure 4.

[1] Chen, Zhe, et al. "Internvl: Scaling up vision foundation models and aligning for generic visual-linguistic tasks." Proceedings of the IEEE/CVF Conference on Computer Vision and Pattern Recognition. 2024.

[2] Chen, Ling-Hao, et al. "MotionLLM: Understanding Human Behaviors from Human Motions and Videos." arXiv preprint arXiv:2405.20340 (2024).

[3] Liu, Haotian, et al. "Visual instruction tuning." Advances in neural information processing systems 36 (2024).

**Limitations:**

The limitations have been discussed in the paper.

---

> ### Author Rebuttal · Authors · 2024-08-07
>
> Thanks for your comprehensive review and constructive comments. We are grateful for your recognition of the novelty of the proposed approach and the importance of our work. In the following, we address your concerns on the off-the-shelf motion estimation and motion editing models, as well as the questions related to experimental details.
>
> **The Limitation of Motion Representation**
>
> We acknowledge the difficulties of obtaining precise motion in real life. However, we found that existing motion estimation algorithms enable us to obtain usable motion sequences in most cases. To verify whether the current pipeline can be applied to real life, we conduct the following experiment.
>
> We invited two participants, one acting as a coach and the other as a trainee. The trainee first performed a source motion sequence. Then, the coach was tasked with generating a target motion sequence that differed from the source sequence. We utilized a pose estimation algorithm (WHAM [1]) to extract these motion sequences and use our method to generate corrective instructions. The trainee is then required to correct his motion based on the corrective instructions. We present an example in Figure 1 of the global response pdf. In this example, it is evident that existing motion estimation algorithms can accurately estimate the motions of both the trainee and the coach. Furthermore, our algorithm is capable of understanding these motion sequences to provide appropriate corrective instructions.
>
> [1] Shin, Soyong, et al. "Wham: Reconstructing world-grounded humans with accurate 3d motion." Proceedings of the IEEE/CVF Conference on Computer Vision and Pattern Recognition. 2024.
>
> **Comparision with VLMs**
>
> Using the above experiment, we also compare our method with Video-Llava and MotionLLM, which can be directly utilized to analyze the videos, and the result is shown in Figure 1 in the global pdf.
>
> The result demonstrates that our method is capable of understanding estimated motion sequences and generating corresponding corrective instructions. In contrast, Video-Llava and MotionLLM struggle to discern the differences in actions between two distinct videos, making it difficult for them to provide appropriate corrective instructions.
>
> **Comparison with Other LLMs**
>
> We appreciate your comment regarding in-context learning. To promote an in-depth comparison, we have fine-tuned various large language models (LLMs) using LoRA. We present the results in Table 1 in the global response PDF.
>
> As observed, our method still shows superior performance than the other baselines. We will include it in the revision.
>
> **W2, Q3: Data Quality**
>
> Thanks for the comment and question. We echo with the concern on the capability of Motion Diffusion Model (MDM), however, we found that the motion generated by MDM is generally good. Furthermore, to enhance its robustness, our proposed data generation pipeline enforces precise alignment between corrective instructions and the intended edits by a carefully designed prompt. For example, by clearly specifying the range of joints to be modified according to the task and efficiently formatting the output that GPT-4 generates as the corrective instructions. These techniques collectively facilitate the creation of natural motions, as evidenced by the figures and videos provided.
>
> **Continuous Representations**
>
> Thank you for your suggestion. Following the approach used in MotionLLM, we conducted the following experiment: First, we utilized a VQ-VAE to map the motion into a (T/4, 512)-dimensional feature, where T is the motion length. Then, we used a linear layer to map this feature to a (T/4, 4096)--dimensional feature. This new feature is directly injected into Llama 3 as the motion embedding. During the fine-tuning process, we update both the parameters of the linear layer and the original weights in the LLMs. The results are reported in Table 5 in the global response PDF.
>
> As observed, our current method still outperforms the continuous baseline across all metrics, confirming that, in the current setting, the features learned by VQ-VAE are not well-suited for the corrective instruction generation task.
>
> **W2, Q5: Generalizable Ability**
>
> To verify the generalization capabilities of our method, we conducted two sets of experiments. First, we presented the results using PriorMdM for validation in Table 2. In this experiment, our algorithm achieved MPJPE and FID scores that are comparable to the Ground Truth and superior to other baselines. Additionally, we validated the generalization of our method on the Fit3D dataset, as shown in Table 3. And the results demonstrate that our method's performance still holds after switching the dataset. These new evidences further verify that the proposed data collection and the training framework can help alleviate the potential bias of motion generation models. The phenomenon observed in Figure 4 is also discussed in the following.
>
> **Q5: Overfitting Phenomenon Shown in Figure 4**
>
> In Figure 4, we presented some corrective instructions and reconstructed motions for different methods. Although the reconstructed motion generated by our method closely resembles the ground truth, we do not consider this to be overfitting after a careful inspection. We brief the reasoning in the following.
>
> First, our task objective is to ensure that the reconstructed motion closely approximates the ground-truth motion. This is important if the proposed is applied to generate personalized corrective instructions (i.e., for a specific person or motion editor). Second, we observe that the instructions generated by our method do not fit exactly to the ground-truth corrective instructions, indicating that there is not a severe overfitting in the generated text.
>
> Again, we are grateful for the constructive feedback, which helps refine our submission. Please feel free to let us know if you have more questions or need more information that will help improve the final rating of our work.

---

> > ### Comment · Reviewer_bYSe · 2024-08-12
> > **Response to rebuttal**
> >
> > Thanks for the author's detailed response. I really appreciate the authors' effort to answer my questions. The authors solve most of my problems. The model architecture can be seen in other models, such as [1]. MotionLLM still can input motion instead of videos. However, this paper still introduces an interesting and useful task. Thus, I decided to increase my rating.
> >
> > [1] Chen, Ling-Hao, et al. "MotionLLM: Understanding Human Behaviors from Human Motions and Videos." arXiv preprint arXiv:2405.20340 (2024).

---

> > > ### Author Response · Authors · 2024-08-12
> > >
> > > Thank you for taking the time to reevaluate our work and for your updated review. We appreciate your acknowledgment of our efforts in addressing your concerns and are glad to know that we have been able to resolve most of your questions. It is encouraging to see that our paper has contributed an interesting and useful task to the field, as you mentioned. We are committed to refining our work (e.g., by incorporating a discussion regarding MotionLLM) further based on the valuable input from you. Once again, thank you for your time and your decision to raise the score.

---

### Official Review · Reviewer_cTdo · 2024-07-11

**Soundness:** 2
**Presentation:** 3
**Contribution:** 3
**Rating:** 5
**Confidence:** 4

**Summary:**

This paper introduces an innovative method designed to generate corrective instructional text, guiding users to achieve desired motions. The approach utilizes existing frameworks to create a dataset of motion pairs and corresponding corrective texts. A new motion-language model is introduced, efficiently generating corrective instructions based on the datasets.

By fine-tuning large language models with these motion-text paired datasets, the method demonstrates precision in instructional generation. Both qualitative and quantitative analyses experimentally confirm the superiority of this approach over traditional methods, which often require extensive manual annotations and struggle with dynamic motions. The researchers’ contributions include a novel data collection pipeline and the application of advanced language models to effectively bridge the gap between motion understanding and natural language instruction generation.

**Strengths:**

1.	The paper introduces a novel task—generating motion corrective instructions—which is both interesting and potentially impactful.
2.	The authors compile a corresponding dataset (it is unclear if the authors plan to make this dataset publicly available).
3.	The writing is clear, and the method is easy to follow.

**Weaknesses:**

1. If I understand correctly, the authors rely on GPT-4 to automatically generate instructions and use MDM to obtain modified motions. How is the quality of this collected data evaluated? Due to the hallucination in large models, the generated instructions may not always be reasonable. Additionally, as a classical method, MDM may not produce natural and controllable motion quality in some corner cases. How do the authors ensure the generation quality of their pipeline?
2. Is the in-context method used by the authors to score the model fair and reasonable? Given the input context length limitation, the number of examples that can be provided to the large model is restricted.
3. Does the authors' division based on upper and lower body introduce any bias? For instance, the instruction "try to touch your toes with your fingertips" cannot be generated.
4. Can the authors provide detailed experimental settings for Llama-3-8B-LoRA? From the results in Table 1, the fine-tuning results based on LoRA are even less effective than the original Llama-3. Do the authors have any explanations or analyses?

**Questions:**

The task proposed by the authors is innovative. My primary concerns are based on the weaknesses. Please address the following in the rebuttal: the quality of the pipeline's generation, comparison strategies with baselines, prompt settings, and experimental setups. This will help illustrate the value of the authors' method.

**Limitations:**

The authors' method does not address social impact.

---

> ### Author Rebuttal · Authors · 2024-08-06
>
> Thank you for your valuable feedback. We appreciate your supportive acknowledgment of the writing quality, the originality of the proposed task with potential impact, the effectiveness of the proposed model, and the novelty in the data collection pipeline. Also, thanks for the summary of the main points that we need to address to help with your evaluation, i.e., the quality of the pipeline's generation, comparison strategies with baselines, prompt settings, and experimental setups. In the following, we provide more details and explanations regarding these points.
>
> **Dataset Access**
>
> Yes, we will release the dataset and the dataset generation pipeline for future research.
>
> **Dataset Quality**
>
> To ensure the quality of the generated corrective instructions by GPT-4, we conducted a thorough manual check of them before performing any experiments and verified that these generated instructions were not influenced by large models' hallucinations.
>
> Regarding the Motion Diffusion Model (MDM), in general, we observe that it exhibits robust motion generation capabilities during our experiment. To further enhance its robustness, when generating corrective instructions, the proposed data generation pipeline ensures that the corrective instructions are precisely aligned with the intended edits, e.g., by specifying the range of joints to be modified according to the task, as well as providing efficient formatting for the generated corrective instructions. Together, they facilitate the creation of natural motions, as evidenced in the provided figures and videos.
>
> **Data Bias**
>
> We would like to clarify that dividing instructions into upper and lower body parts does not limit the complexity of the motion to be generated. For example, the aforementioned target motion can be addressed by defining modification joints more specifically, i.e., we can instruct GPT-4 to generate instructions that modify both the right hand and left leg to produce a sequence that signifies ``try to touch your toes with your fingertips.''
>
> **Comparison with Other LLMs Using LoRA**
>
> We appreciate your point regarding in-context learning and fairness. To facilitate a more direct comparison, we fine-tune various large language models (LLMs) based on LoRA. The results are presented below.
>
> | Method | BLEU ⬆ | ROUGE ⬆ | METERO ⬆ | CLIPScore ⬆ | MPJPE ⬇ | FID ⬇ |
> | - | - | - | - | - | - | - |
> | Llama-3-8B-LoRA | 0.11 | 0.17 | 0.36 | 0.78 | 0.37 | 5.03 |
> | Mistral-7B-LoRA | 0.13 | 0.17 | 0.36 | 0.79 | 0.30 | 5.02 |
> | Ours | **0.14** | **0.27** | **0.47** | **0.80** | **0.21** | **4.52** |
>
>
> As observed, our method still outperforms the baselines that have been fine-tuned using LoRA, which we will add to the revised version for an improved understanding of the efficiency.
>
> **Detailed Experimental Settings for Llama-3-8B-LoRA**
>
> We fine-tuned Llama3 following the methodology described in [1]. Specifically, we converted motion tokens into text and combined them with the template as input for the network while conducting the fine-tuning using LoRA. During training, we set the learning rate to 1e-4, the LoRA rank to 16, and the batch size to 512. The entire fine-tuning process lasted for 6,000 training steps.
>
> We also would like to make some clarifications about the experimental results. After fine-tuning Llama3 with LoRA, the generated corrective instructions can induce higher-quality reconstructed motions (FID: 2.09) compared to the original Llama-3 before fine-tuning (FID: 3.04). However, we observed that after fine-tuning using LoRA, the output of Llama3 became more varied, e.g., the instructions generated on the test set exhibited greater variance compared to those generated by the in-context learning baseline with the original Llama-3. This evidenced that simply fine-tuning Llama-3 using the generated data would not result in a satisfactory corrective instruction generation, e.g., due to overfitting or catastrophic forgetting. Although the outputs can induce similar target motion sequences compared to the ground truth, the increased variance in the text can lead to a decrease in the overall NLP metrics such as BLEU, ROUGE, and METERO.
>
> [1] Zhang, Yaqi, et al. "Motiongpt: Finetuned llms are general-purpose motion generators." Proceedings of the AAAI Conference on Artificial Intelligence. Vol. 38. No. 7. 2024.
>
>
> In light of these clarifications, we hope to have addressed the concerns raised and further justified the approach and focus of our study. We hope these clarifications have addressed the concerns raised and further justified our study’s approach and focus.  Please feel free to let us know if you have more questions or need further information that will help improve the final rating of our work.

---

> > ### Comment · Reviewer_cTdo · 2024-08-11
> >
> > I appreciate the authors for their expert, comprehensive, and detailed analysis. The authors have thoroughly addressed the quality of their dataset and the rationale behind their experiments in the rebuttal. I also reviewed the comments from other reviewers and the authors’ corresponding responses. The authors’ motivation is both interesting and reasonable. Although their pipeline largely builds on previous methods, it still demonstrates potential application value, which is substantiated by the experimental analysis provided in the rebuttal. Therefore, I am inclined to raise my score.

---

> > > ### Author Response · Authors · 2024-08-11
> > > **Thanks for your response**
> > >
> > > We would like to express our gratitude for your time, effort, and detailed evaluation of our paper. We appreciate your acknowledgment of the comprehensive analysis provided in our rebuttal, as well as your recognition of the potential application value of our research. We are glad that our clarifications and explanations have addressed your concerns, and we are thankful for your decision to raise the score.

---

### Official Review · Reviewer_cjte · 2024-07-13

**Soundness:** 3
**Presentation:** 3
**Contribution:** 2
**Rating:** 5
**Confidence:** 4

**Summary:**

This paper works on generating corrective instructional text from source and target human motions. Specifically, it utilized existing motion editing framework to collect a dataset for this task. Then, an LLM instruction generation method was proposed to generate text from source and target motions. This paper also shows better performances than baseline methods.

**Strengths:**

1. This paper is well-written and easy to follow, and the supplementary video effectively enhances the content.
2. The motivation of generating corrective instruction from source and target human motions is very interesting and has potential in real-world applications.
3. The proposed method shows better performance than previous methods in the evaluation.

**Weaknesses:**

1. The overall technical contribution is limited. The presented method seems to be an extended application of existing text-to-motion methods, and the adopted components are basically from previous methods.

2. Some recent works (e.g., [1]) are missing to discuss or compare.
[1] AvatarGPT: All-in-One Framework for Motion Understanding Planning Generation and Beyond, CVPR 2024

3. MotionGPT used T5-770M as its LLM backbone, while the proposed method uses ChatGPT-4, so a direct comparison with MotionGPT using different LLMs doesn't seem to be fair.

**Questions:**

To resolve my concern regarding the technical contribution. I would like to know what are the main challenges of this addressed task compared to existing text-to-motion tasks?

**Limitations:**

Limitations have been discussed in Section 5.

---

> ### Author Rebuttal · Authors · 2024-08-05
>
> Thank you for the constructive suggestions. We also appreciate your acknowledgment of the writing quality, the novelty and potential for real-world applications, and the effectiveness of the proposed method. However, it seems that the technical difficulty as well as the technical contributions of our work may not have been entirely clear. In the following, we address these concerns.
>
> **Main Challenges and Technical Contribution**
>
> Thanks for the questions. To alleviate the impression that the proposed method seems to be an extended application of text-to-motion methods, we summarize the primary challenges regarding corrective instruction generation given existing works:
>
> + Lack of Training Data
>
> Unlike text-to-motion or human pose correction (which can be annotated through simple pipelines [1]), human motion sequences involve temporal changes. Annotating the differences between these temporal changes (for corrective instruction generation) is challenging, and, to the best of our knowledge, no existing datasets provide such annotations to enable training.
>
> To resolve this issue, we contribute by proposing a novel prompting technique and a method that generates editor-aware instructions so that the generated motion sequences can faithfully align with the corrective instructions. This process is highly scalable and can be extended to different motion editors and motion editing tasks, significantly facilitating the training of corrective instruction generation models.
>
> + Learning Complexity Beyond Text-to-Motion/Motion-to-Text Annotation
>
> Previous methods for training text-to-motion models involve either using an existing vocabulary for motion tokens or assigning new learnable embeddings, followed by fine-tuning with techniques like LoRA. We tried both approaches, but the results of utilizing one of them alone were not satisfying. There are two main reasons: First, using a fixed vocabulary and embeddings prevents capturing the correlation of motion differences and corrective instructions, as the weights are trained on tasks with a large domain gap. Second, while new embeddings can be learned with LoRA, the distribution of the original vocabulary's embeddings imposes constraints, making the learned embeddings suboptimal, especially given the smaller scale of training data for corrective instructions.
>
> To address these challenges, we integrate the goods of both. We use existing vocabulary tokens for their rich semantics and fine-tune all embeddings to maximize performance and reduce the domain gap. We also introduce an anchor loss to prevent the embeddings from diverging. These training techniques prove effective, resulting in superior performance.
>
> + Selective Expression of Differences in Specific Activity
>
> In real-world applications, selectively expressing differences between motion sequences is crucial. For example, as kicking balls, slight changes in the contact point between the foot and the ball can significantly impact performance, while arm swing amplitude may not. Capturing relevant details for effective corrective instruction generation is challenging. Therefore, we can contribute by sharing the data generation and training pipeline to aid in investigating this important issue.
>
> These challenges highlight the differences between the proposed task and the text-to-motion and motion-to-text paradigms. The solutions presented also signify technical contributions to addressing these problems. Thank you for your feedback; we will revise our manuscript to provide a clearer contrast.
>
> [1] Delmas, Ginger, et al. "Posefix: correcting 3D human poses with natural language." Proceedings of the IEEE/CVF International Conference on Computer Vision. 2023.
>
> **Comparison to AvatarGPT**
>
> Thanks for your suggestion, we will discuss AvatarGPT in our revision, and sorry for not doing so due to the delayed release of accepted papers in CVPR 2024. We agreed that AvatarGPT is a compelling work expanding the range of text that can be comprehended by text-to-motion methods, enabling the generation of complex and long motion sequences from text. Additionally, like our work, AvatarGPT introduces an efficient data generation process (but not for our task), benefiting the development of text-to-motion models.
>
> In contrast to AvatarGPT, our focus is corrective instruction generation, where the inputs are two motion sequences (instead of text and motion), and the output is the text describing the differences between the input motion sequences (but not the edited motion as in text-to-motion generation). The generated text shall efficiently inform a text-based motion editing model to change the source motion sequence towards the target (reference) motion sequence to facilitate tasks like motion re-targeting and sports coaching. This fundamental difference in input-output dynamics makes it difficult to directly compare our work with AvatarGPT.
>
> We will include this discussion about AvatarGPT in our revision.
>
> **Comparison using T5**
>
> Thanks for the suggestion. First, we would like to correct a typo: in our experiments, the backbone we have used is Llama-3 but not GPT-4. We acknowledge your concern about the impact of different backbones on the performance. To resolve this issue and test the robustness of our method, we have fine-tuned T5-770M, as in Motion-GPT, to provide a more direct comparison with MotionGPT. The results are reported in Table 4 in the global response pdf.
>
> Based on the new results, we can observe that even though a smaller backbone is used (T5) as our LLM backbone, our method still outperforms MotionGPT, demonstrating its effectiveness and robustness across different backbones.
>
> We are grateful for the constructive feedback, which is instrumental in enhancing the quality and clarity of our submission. We hope that the discussions and proposed revisions can help address the current concerns. Please feel free to let us know if you have more questions that can help improve the final rating of our work.

---

> > ### Comment · Reviewer_cjte · 2024-08-12
> >
> > I appreciate the author's feedback and believe the rebuttal has addressed my concerns. I have also considered the issues raised by other reviewers and noted that the rebuttal provided positive responses and clear clarifications for those as well. As a result, I am inclined to increase my score.

---

> > > ### Author Response · Authors · 2024-08-13
> > >
> > > Thank you for taking the time to reevaluate our work and for providing your updated feedback. We are glad that our rebuttal has successfully addressed your concerns and clarified the key contributions of our research. Your acknowledgment is highly appreciated and serves as motivation for us to continue refining our manuscript. We will ensure that all necessary revisions are incorporated in the final version. Once again, we express our gratitude for your thorough review and your openness to adjusting your assessment. Your constructive feedback has been invaluable in enhancing the quality of our work.

---

### Official Review · Reviewer_A1J6 · 2024-07-15

**Soundness:** 3
**Presentation:** 3
**Contribution:** 2
**Rating:** 5
**Confidence:** 5

**Summary:**

This paper presents CigTime for generating motion corrective instructions. The key idea is to leverage motion editing to create datasets of motion triplets and use a fine-tuned language model to generate precise and actionable instructions.

**Strengths:**

(1) Introduces a approach for converting motion discrepancies into actionable textual guidance, with applications in sports coaching, rehabilitation, and motor skill learning.

(2) Proposes a motion-editing-based pipeline to efficiently generate large datasets, reducing the dependency on manual annotations.

(3) Demonstrates the effectiveness of the method through comprehensive evaluations, outperforming existing models.

**Weaknesses:**

(1) For prompting ChatGPT-4 for corrective instruction generation like in Fig.2, I think text description for upper/lower body motion is not enough, prompting ChatGPT4 to give details on how to achive the desired motion is need for "corrective instruction".

(2) For corrective motion, blending source and target motion could generate not human-like motion, why don't you try to directly generate corrective motion by masking source motion with editing text prompt.

(3) Although corrective motion instruction generation is helpful for lots of applications, I believe the proposed method don't show its potential to enhance any application, just like a toy-level verification.

(4) For LLM finetuning, do you extend text vocabulary, or just use a number(index) to represent motion token and directly put it into LLM?

(5) For comparison with ICL of other LLMs, it's not fair. The motion index list space is so large and LLM don't see possible mapping between it and text in pretraining, LLM could only succeed when you put similar index list and get similar text like few-shot example. When you try some different index list, it's likely to output similar text. This is because the mapping between index list and text is not established using few-shot examples. So maybe you should provide all these LLM trained with a PEFT(like LoRA) method.

(6) What is the result on KIT? And to support finger motion, Motion-X is an optional dataset.

**Questions:**

(1) For tokenizer training, it lacks the loss for codebook.

(2) In line 445, FID from 1.44 to 0.24 is not subtle if you are fimiliar with t2m task. Also, why is FID for Fit3D better than HumanML3D? It's quite different from the changing of other metrics.

**Limitations:**

As described in questions and weakness.

---

> ### Author Rebuttal · Authors · 2024-08-06
>
> Thank you for your valuable feedback. We appreciate your positive comments on the proposed approach and its application potential, data generation efficiency, and the effectiveness of the method. Below, we address your questions regarding corrective instruction details, motion generation, practical application demonstrations, fine-tuning, and additional results.
>
> **More Advanced Data Generation Prompts**
>
> Thanks for the suggestion. To achieve more precise and detailed instructions that illustrate "how", we have prompted GPT-4 to generate the following corrective instructions:
>
> [Raise the left hand 20cm higher by changing the elbow and shoulder joints];
>
> [Turn your head to the right by 30 degrees more by changing the neck joint].
>
> These prompts can provide more detailed (localized) information on how to achieve the target motion. However, due to limitations in the motion editor, for example, the detailed instructions will result in unnatural movements in the current format, when compared to using more global corrective instructions in terms of upper or lower body instead of prescriptive local descriptions.
>
> **Corrective Motion Generation**
>
> Our current data generation process aligns with the procedure you proposed. Specifically, during the target motion generation process with the Motion Diffusion Model (MDM), we mask the source motion and modify the unmasked parts corresponding to the lower or upper body according to the corrective instructions. We will clarify this point in our revised paper.
>
> **Application**
>
> To demonstrate the potential of our method to enhance applications, we conducted the following experiment.
>
> We invite two participants, one acting as a coach and the other as a trainee. The trainee first performs a source motion sequence. Then, the coach is tasked with generating a target motion sequence that differs from the source sequence. We utilize a pose estimation algorithm (WHAM [1]) to extract these motion sequences and use our method to generate corrective instructions. The trainee is required to correct his motion based on the corrective instructions.
>
> We compare our method with Video-Llava and MotionLLM, which can be directly utilized to analyze videos. We present an example in Figure 1 of the global response.
>
> The result demonstrates that our method can understand low-quality (as compared to motion capture) motion derived from pose estimation algorithms and generate corrective instructions. In contrast, Video-Llama and MotionLLM struggle to discern the differences in actions between two distinct videos, making it difficult to provide appropriate corrective instructions. We will enhance our analysis demonstrating the practicality of our approach for real-world applications.
>
> **Extend Text Vocabulary**
>
> Currently, we use indices to represent motion tokens. However, we further conducted the following experiment to extend the text vocabulary. We present the results below.
>
> | Method | BLEU ⬆ | ROUGE ⬆ | METERO ⬆ | CLIPScore ⬆ | MPJPE ⬇ | FID ⬇ |
> | - | - | - | - | - | - | - |
> | Llama-3-8B-Extended | 0.12 | 0.23 | 0.44 | 0.80 | 0.27 | 5.43 |
> | Mistral-7B-Extended | 0.18 | 0.27 | 0.42 | 0.81 | 0.19 | 1.45 |
> | Ours-Extended | **0.24** | **0.37** | **0.55** | **0.84** | 0.16 | 1.50 |
> | Ours | **0.24** | 0.35 | 0.52 | 0.82 | **0.13** | **1.44** |
>
> We find that employing an extended vocabulary can enhance the text-based metrics which are calculated based on the generated instruction and ground truth instructions. However, these instructions cause a decline in the motion editing performance, resulting in a reduction in MPJPE and FID. From the perspective of the task definition, we require a model that prioritizes high reconstruction quality over instruction quality. Therefore, extending vocabulary is more detrimental than beneficial for our task, and we will discuss in the revised version.
>
> **Compare to Other LLMs with LoRA**
>
> We appreciate your observations regarding in-context learning. We have fine-tuned various large language models (LLMs) using LoRA. The results are presented in Table 1 in the global response pdf.
>
> As observed, our model still outperforms the other baselines, which demonstrates the effectiveness of the proposed training pipeline.
>
> **Experiments on KIT**
>
> We fine-tune and evaluate our method and baselines on the KIT dataset. We report the results in Table 3 in the global response pdf.
>
> On the KIT dataset, our method still outperforms other baselines across all metrics, demonstrating the generalization capability.
>
> Extension to fingers also requires effort in improving the motion editing models, for which we resort to future work.
>
> **Loss for Codebook**
>
> Following [2], we apply exponential moving averages (EMA) to update the codebook to stabilize the training process as we mentioned in Line 185, which helps alleviate the need to utilize codebook loss for tokenizer training.
>
> **FID Metrics**
>
> We greatly appreciate your detailed review and attention to the specifics of our manuscript. Upon re-checking our supplementary material, we discovered an error: the FID score on Fit3D should have been reported as 1.24, not 0.24 as previously stated. The commentary in line 445 regarding the subtlety of changes in CLIP score, MPJPE, and FID was indeed based on this correct result. The other numbers are correct after a double-check. We apologize for any confusion caused by this error and are grateful for your carefulness.
>
> Once again, we thank you for your constructive feedback and hope that our rebuttal addresses your questions. Please feel free to let us know if you have more questions that can help improve the final rating of our work.
>
> [1] Shin, Soyong, et al. "Wham: Reconstructing world-grounded humans with accurate 3d motion." IEEE/CVF Conference on Computer Vision and Pattern Recognition. 2024.
>
> [2] Van Den Oord, Aaron, and Oriol Vinyals. "Neural discrete representation learning." Advances in neural information processing systems (2017).

---

### Author Rebuttal · Authors · 2024-08-07

We would like to express our gratitude for the reviewers' careful evaluations and constructive feedback on our manuscript.

In response to the reviewers' feedback, we appreciate their recognition of the novelty in the task/approach presented, for generating motion corrective instructions (R2, R3, R4). As noted, the key strength of our method lies in its innovative use of motion editing to convert motion discrepancies into precise, actionable textual guidance. This method, with potential applications in sports coaching, rehabilitation, and motor skill learning, is an important contribution to the field (R1, R2, R3, R4).

The reviewers also acknowledged our efficient motion-editing-based pipeline for generating large datasets, thereby reducing dependency on manual annotations. This innovation has been well received for its effectiveness as demonstrated through comprehensive evaluations, where it outperformed existing models (R1, R2, R3).

We are further encouraged by the positive feedback regarding the readability of our paper and the supplementary video (R2, R3), and the acknowledgment of our method's superior performance over previous methods in evaluations (R1, R2, R3).

These help underscore the contributions our work aims to make, and we are delighted that they resonate with the reviewers.

The reviewers have also raised concerns or questions that we will address in our rebuttal, which include:

1. Questions regarding the specifics of the generated corrective instructions.
2. Clarifications on the motion generation process and its efficiency.
3. Requests for additional examples and demonstrations of the method in real-world scenarios.
4. Clarification of the technical difficulty, as well as the fine-tuning process.
5. Requests for further results and comparisons with baselines.
6. Questions related to the use of existing motion estimation and editing models, and their impact on the proposed approach.

In our rebuttal, we will provide more details and explanations to address these concerns and clarify any misunderstandings, ensuring that our work's technical contributions and the effectiveness of the proposed method are well understood. We are also committed to addressing each of them in the revision.

---

### Decision · Program_Chairs · 2024-09-25

**Decision:**

Accept (poster)

**Comment:**

This paper presents a novel approach to generating corrective instructional text for human motions. The authors propose a method that leverages existing motion editing and generation models to create datasets of motion triplets (source, target, and corrective text), which are then used to fine-tune a large language model for generating precise and actionable corrective instructions.

The reviewers initially provided mixed feedback, with scores ranging from "Borderline accept" to "Weak Accept." However, it's worth noting that three out of the four reviewers are first-time reviewers for NeurIPS, which may have influenced the initial diversity in assessments. The authors' response to the reviewers' concerns was particularly noteworthy. They provided detailed explanations, additional experiments, and clarifications that effectively addressed most of the raised issues. As a result, three out of four reviewers were convinced to raise their scores after the rebuttal. This demonstrates the authors' strong engagement with the review process and their ability to effectively communicate the value and novelty of their work.

The key strengths highlighted by the reviewers include the novelty of the task and approach, with potential applications in sports coaching and motor skill learning; the efficient data generation pipeline that reduces dependency on manual annotations; and the effectiveness of the method, as demonstrated through comprehensive evaluations. Some concerns were initially raised about the technical contribution and comparison with existing methods, the quality of the generated data and potential overfitting, and the fairness of comparisons with baseline methods. However, the authors successfully addressed most of these concerns in their rebuttal, providing additional experiments, clarifications on their methodology, and more comprehensive comparisons.

As the Area Chair, I have also carefully read through the paper and agree that it is above the acceptance threshold for NeurIPS. The authors have demonstrated a novel approach to a challenging problem, and their method shows promise for real-world applications. Their successful engagement with reviewers and ability to address concerns effectively further strengthens the case for acceptance.

I recommend accepting this paper for publication at NeurIPS. However, I strongly encourage the authors to incorporate the clarifications and additional results provided in their rebuttal into the camera-ready version of the paper. This will ensure that the final publication addresses the reviewers' questions and presents a more comprehensive view of the work's contributions and limitations.